# Increasing aridity threatens the sexual regeneration of *Quercus ilex* (holm oak) in Mediterranean ecosystems

**Patricio Garcia-Fayos**[1]*, **Vicente J. Monleon**[2], **Tiscar Espigares**[3], **Jose M. Nicolau**[4], **Esther Bochet**[1]

**1** Desertification Research Centre (CIDE, CSIC-UV-GV), Moncada, Valencia, Spain, **2** US Forest Service Pacific Northwest Research Station, Corvallis, Oregon, United States of America, **3** Department of Life Sciences, Faculty of Sciences, University of Alcalá, Alcalá de Henares, Madrid, Spain, **4** Technical School and Environmental Sciences Institute, University of Zaragoza, Huesca, Spain

* patricio.garcia-fayos@ext.uv.es

## Abstract

Knowledge of the recruitment of dominant forest species is a key aspect for forest conservation and the ecosystem services they provide. In this paper, we address how the simultaneous action of climate change and the intensity of land use in the past influence the recruitment of a forest species that depends on the provision of nurse plants to recruit. We compared the number of saplings (up to 15 years old) and juveniles (16 to 50 years old) of *Quercus ilex* in 17, 5.3 ha plots in the Iberian System (eastern Spain). We used a gradient of past deforestation intensity crossed with two levels of average annual precipitation, one of them at the lower limit of the species' precipitation niche (semi-arid) and the other at the optimum (sub-humid). We also examined the association between recruits and nurse plants and the effect on this association of plot-scale factors, such as seed abundance (reproductive *Q. ilex*), microsites (nurse species and soil availability), and large herbivores. The increase in aridity in the last decades has drastically reduced the recruitment of new individuals in the forests of *Q. ilex* located in the lower limit of their precipitation niche, regardless of the intensity of past deforestation that they suffered. Recruitment in these climatic conditions depends almost exclusively on large trees and shrubs whose abundance may also be limited by aridity. The lack of regeneration questions the future of these populations, as the number of individuals will decrease over time despite the strong resistance of adult trees to disturbance and drought.

## Introduction

Climate projections indicate that the Mediterranean is one of the regions most vulnerable to global change [1]. The western Mediterranean is experiencing two large-scale processes that can have profound influence in the dynamics of its forest ecosystems. While humans have affected the landscapes of this region for millennia, there has been a sharp decline in human population in rural areas starting on the second half of the 20<sup>th</sup> century, which has resulted in

**Data Availability Statement:** The data files have been deposited in the Figshare digital repository: https://doi.org/10.6084/m9.figshare.12855170.

**Funding:** This work was supported by Ministerio de Economía, Industria y Competitividad, Gobierno

de España, CGL2013-42213-R, to Dr Esther
Bochet; Ministerio de Ciencia, Innovación y
Universidades and European Regional
Development Fund, RTI2018-095037-B-I00 to EB;
Conselleria d'Educació, Investigació, Cultura i
Esport, PROMETEO 2016/021 to PG-F. The CSIC
Open Access Publication Support Initiative gave
support of the publication fee through its Unit of
Information Resources for Research (URICI). The
funders had no role in study design, data collection
and analysis, decision to publish, or preparation of
the manuscript.

**Competing interests:** The authors declare that no
competing interests exist

the recovery of vegetation and increase of forest area [2]. Concurrently, mean annual temperature has increased at an estimated rate of 0.30 ˚C per decade [3] and precipitation decreased at a rate of 18.7 mm per decade, leading to an intensification of aridity (+24.4 mm of annual evapotranspiration per decade) and drought severity [4]. Since changes in both human pressure and climatic conditions have occurred simultaneously, it is imperative to understand how those two processes and their interaction influence forest dynamics and, specifically, regeneration.

*Quercus ilex* L. is an evergreen oak that constitutes a keystone species for biodiversity and ecosystem services in western Mediterranean countries [5]. Their great ability to sprout after fires, logging, browsing and droughts has allowed their populations to persist for a long time despite intense human activity [6]. Like other *Quercus* species, it is a renowned seed mast species [7]. Acorns are an important food resource for many animals, so after falling to the ground, those that are hidden or stored and not consumed by their dispersers are the ones that are more likely to escape predation [8, 9]. Seeds, but also seedlings, are very sensitive to dehydration and rely mostly on shelter against desiccation to survive and establish [10, 11]. In *Q. ilex*, recruitment occurs in full sun only when annual precipitation is 700 mm or more [10], but as precipitation decreases, seedlings need shade to compensate for evapotranspiration, until reaching a lower limit of annual precipitation, around 250 mm, when the water deficit can no longer be compensated by the shade [10, 12]. In the wild, nurse plants facilitate the establishment of seedlings in many *Quercus* species under Mediterranean climate conditions because they improve soil conditions, reduce predation on seeds and seedlings, and provide shade that favors a positive water balance of seeds and young plants [8, 13–18]. However, *Quercus* seedlings need light to grow, so the intense shade that favors recruitment under high evaporative demand makes their subsequent development to seedlings and juveniles difficult or even impossible [19, 20].

While the scale of the interactions between the nurse and the facilitated plant is local [21], the strength of this relationship may be affected by factors operating at larger scales, such as topography, climate, seed availability, abundance of seed dispersers, herbivore pressure, and abundance of microsites favoring germination and survival (i.e. nurse plants) [11, 13, 22–25]. Moreover, all those factors not only can affect seedling establishment and survival directly, but they can also interact with the effect of the nurses.

In this study, we analyse data on *Q. ilex* plants that established over the last 50 years in forest remnants dominated by this species in the Iberian Range (Eastern Spain). This period corresponds with a very intense rural exodus followed by a recovery of the vegetation cover [26], but also with an increase in temperature and aridity [4]. We hypothesized that both intense past deforestation and increasing aridity during forest recovery would impose restrictions on *Q. ilex* recruitment, and that these restrictions would be greater in areas where the species was on the edge of the precipitation niche. To test these hypotheses, we compared the recruitment in sites selected along a deforestation gradient with contrasting mean annual precipitation, while keeping all other environmental and human use factors similar. Mean annual precipitation at the beginning of the recruitment period considered here was at the lower limit for the species range in some of these sites and close to the mode of the distribution in the rest [27]. We used cover of *Q. ilex* trees as a surrogate of human-caused deforestation, with a deforestation intensity gradient that goes from plots with a single tree to plots with the highest possible tree cover of the species under the climatic conditions of the area where the plot is located.

In addition, we explored the association of *Q. ilex* recruits to individuals of plant species potentially acting as nurses. We first examined the spatial association between recruits and potential nurse species at the microscale. Then, we examined the effect of several factors at a broader scale, including precipitation and stand attributes (past deforestation intensity,

abundance of reproductive *Q. ilex*, abundance of nurse plants, soil availability and herbivory pressure) on the strength of this association.

## Material and methods

### Plot selection and characteristics

We selected 17 plots of 231 x 231 m in a 130 km east-west transect in the Iberian Mountain Range from Vilafranca del Cid (Castellón province) to Traid (Guadalajara province) (Eastern Spain). These plots are a subsample of the 138 plots used in a previous study by the authors [28] designed to minimize the variations between sites in the magnitude of the factors that affect the development of *Quercus ilex* that were not those we want to study.

The altitude of the plots ranged between 1060 and 1400 m a.s.l., and all of them were located in high and flat areas with a slope angle less than 6 ° (S1 Table). Parent lithology was massive limestone from the Cretaceous and Jurassic periods and the soil type was Molllic Haploxeralfs. The mean annual temperature varied between 9.8 ° and 12 ˚C and dry and wet years affected similarly all the area during the period 1700–2012 [29]. We visited all the plots to confirm that *Q.ilex* was the dominant tree species and if there were only a few individuals left, we confirmed their dominance in the adjoining plots.

Since we selected plots without signs of having been cultivated, we assumed that the variation in *Q. ilex* cover between plots was an indication of the intensity of deforestation suffered (firewood, wood and feed for domestic livestock). In support of this idea, we previously verified that the distance to the closest human settlement explained the variation in the tree cover of *Q. ilex* in the study area, while the geomorphological characteristics of the landscape, such as slope and aspect that inform on fertility and water availability in soils, they did not [28].

The human population density in the entire study area fell from 20 inhabitants per square meter in 1960 to five in 2015 (Spanish Statistical Office: https://www.ine.es/en/, last consulted on 01 / 02/2020), which led to a decrease in livestock's density of an order of magnitude and an increase of wild herbivore abundance, which did not compensate for this reduction in density (S1 Appendix). These processes allowed for the recovery of forest vegetation in the region [26]. However, we assumed that this recovery would not have modified the observed deforestation pattern, given the insignificant temporal variations in the tree cover of *Q. ilex* between 2004 and 2012 in all plots (<5%, [28]). It is coherent with the slow increase in the growth and density of the forests of this species in the Iberian Peninsula in the period studied [30].

Eleven of the plots located in an area with mean annual precipitation between 400 and 450 mm (semi-arid hereafter) and the other six in an area with mean annual precipitation between 600 and 650 mm (sub-humid hereafter) (S1 Table). We obtained values of mean annual precipitation for each plot from the Digital Climatic Atlas of the Iberian Peninsula [31] (data from 1951–1999 with 200 m spatial resolution). The difference in the number of plots between semi-arid and sub-humid was a consequence of the number of plots available [28] and the very low recruitment density in the semi-arid.

Average daily temperatures in the study area increased 0.54 ˚C (95% CI 0.38 to 0.68) per decade between 1973 and 2005 [32], which in itself it has meant a significant increase in the demand for water by plants and, therefore, an increase in climatic aridity in all the plots [4].

Within each precipitation level, we selected plots to represent the full range of past deforestation values, from the densest *Q. ilex* forests to areas where only one tree remained. Because precipitation influences the maximum possible of tree cover value of *Q. ilex* in each precipitation level, we developed a standardized deforestation index within precipitation levels (S1 Table). This index is the ratio of the difference between the observed maximum *Q. ilex* cover value within a precipitation level and the value in the plot, divided by the maximum observed

value in the region, and takes values from one (treeless areas) to zero (maximum potential tree cover). We obtained the maximum value in each precipitation level from a data set of 374 plots in the semi-arid and 209 plots in the sub-humid that includes all the plots in the studied region in which *Q. ilex* is the dominant species under the climatic conditions considered [28].

We measured stand attributes that can affect recruitment and the strength of the association between nurses and recruits, including precipitation, past deforestation intensity, relative abundance of reproductive *Q. ilex* and nurse plants, herbivory pressure, and soil availability for the acorns to germinate and establish. Within each plot, we established two 231 x 14 m parallel subplots separated 100 m from each other. We estimated the relative cover of reproductive *Q. ilex* and of nurse plants in spring and autumn of 2016, with the line intercept method. We used a 14 m long tape arranged transversally every 10 m along the subplots (a total of 644 m per plot). We defined reproductive *Q. ilex* as trees with presence of acorns or their remains, and those without them but with DBH$\geq$ 20 cm [33]. We defined a potential nurse as plant species whose morphological characteristics (plant size and shape, leaf size and leaf density and layout, presence of thorns and spines, etc.) are attractive to the acorn dispersers or can provide protection to the acorns and seedlings against drying or herbivory (S2 Table). As an indication of soil availability, we estimated the probability that an acorn that reached the ground could germinate and root, that is, where there were no rock or stones in the first 5 centimetres of soil. We estimate this index as the proportion of 500 points, distributed in 10 transects of 10 m each, distributed evenly in the plot, where a metal rod 5 mm in diameter could be introduced into the ground up to 5 cm deep. Since we did not observe direct evidence of herbivory in seeds, seedlings, saplings, juveniles and adults of this species at any of the sites studied, we decided to use indirect evidence to determine herbivory pressure. To obtain an indication of the current pressure of herbivory on the plots, we used the proportion of forty-six 10 x 14 m sections per plot where we could find faecal pellets of domestic or wild herbivores, footprints, wool or hairs attached to bushes and trees.

Permits were not required to sample the study plots since they were not located in areas with restricted access and the type of sampling was neither extractive nor destructive, except in the case of *Q. ilex* individuals necessary to establish age relationships with diameter (see Age determination for further information).

## Recruit sampling and measurement

In the spring and autumn of 2017, we tallied all recruits of *Q. ilex* that originated from sexual reproduction in two 231 x 14 m subplots per plot. We defined a recruit as any *Q. ilex* plant that had a root collar diameter equal to or less than 50 mm and was not connected to any surrounding *Q. ilex* adult by the roots. To avoid tallying vegetative shoots of adult plants, we discarded potential recruits that were one meter or less from the stem of an adult. We excavated the remaining potential recruits and examined their root system carefully until we had full evidence that they were not connected to any adult plant. Sometimes, we found vegetative patches of *Q. ilex* with dozens of stems less than 1.5 m tall, densely sprouting from a common root network, a consequence of repeated logging in the past [6]. These stems generally do not produce acorns and have a greater competitive advantage over recruits that originated from sexual reproduction, because their connection to the mother plant provides them with water and carbohydrates. The possibility of survival of any seedling established under these vegetative patches is negligible [34], and together with the effort it would take to discriminate any recruit of sexual origin in those patches, led us to exclude those areas from sampling. We measured the diameter of the root collar of each recruit with callipers to the nearest 0.01 mm in two perpendicular directions and calculated the mean diameter (see "age determination" section).

To study the association between potential nurse plants and recruitment of *Q. ilex*, we used a retrospective sampling design (microsite-level sampling) [35]. We recorded the presence of individuals of those species potentially acting as nurses within 50 cm around every *Q. ilex* recruit. We also sampled 46 additional random, recruit-free points per plot (every 10 m along the east edge of each subplot) and checked them for the presence of nurse plants in the same way.

## Age determination

Initially, we attempted to determine the age of each recruit from its basal area. However, we were unable to apply this procedure because more than 85% of the recruits had two or more stems. The plants of *Q. ilex* have the potential to produce secondary stems when the primary stem is damaged, and to do it repeatedly after each disturbance [6]. The size of these secondary stems does not reflect the age of the recruit, but only that of the sprout. Since all sprouts originate from the same root system, we built models relating the diameter of the root collar to the age of the plant. We sampled 171 recruits and determined their age from tree-ring counts in the laboratory (see S2 Appendix for details on tree-ring determination and model construction and [36] for a similar age determination procedure). We then used those models to estimate recruit age for all the tallied recruits.

The collection of *Q. ilex* is not subject to any prohibition by laws or regulations of a national, regional or local nature that affect the study territory except for the use of firewood, which, given the size of the individuals sampled for this research, it would not apply.

## Data analyses

To examine the effect of precipitation and past deforestation intensity on the number of *Q. ilex* recruits at the plot level, we fitted Poisson log-linear models. We used precipitation and deforestation intensity as fixed variables and included the interaction term between them to assess whether the effect of past deforestation on the number of recruits was different for the semi-arid and sub-humid sites. We kept the interaction in the final model only when it was significant at the 0.05 level. We used site as a random effect, because the 17 plots where clustered in six different municipalities (S1 Table). Since the effect of deep shade on recruits switches from positive to negative at approximately 15 years of age [19], we grouped the recruits into two age classes: saplings (15 years old or less) and juveniles (between 16 and 50 years old).

To examine the association to nurse plants at the microsite level, and how variables at the stand level modify this effect, we used hierarchical logistic regression models (S3 Appendix). We started with a logistic regression that related the log of the odds of *Q. ilex* recruit presence to the presence of a nurse plant within 50 cm around the recruit or random point. Because of the retrospective nature of the sampling, only the slope of this model has a meaningful interpretation [35]. We included random plot effects for both the intercept and slope parameters. Next, we modelled the slope parameter as a function of the plot-level variables, including precipitation level, past deforestation intensity, relative cover of reproductive *Q. ilex*, relative cover of nurse plants, soil availability, and abundance of herbivores and used the AIC criteria to select the best fitted model.

We used the glmer function (*lme4* package, v 1.1–20 in R 3.6.1) for the analyses.

## Results

### Relationship between root-collar diameter and age

The youngest plant sampled in the semi-arid plots, based on the ring count, was seven years old and had an average root-collar diameter of 3.92 mm. The youngest plant in the sub-humid

plots was two years old and had a diameter of 2.89 mm. Fifty-year-old plants in the semi-arid and sub-humid plots had root-collar diameter of 28.52 and 37.60 mm, respectively. Models relating age to diameter of the root-collar of recruits explained most of the variation of data within each precipitation level, $R^2 = 0.48$ for the semi-arid and $R^2 = 0.65$ for the sub-humid (S2 Appendix).

### Recruit characteristics and age structure

We found 1697 recruits, but their density differed hugely between precipitation levels. The maximum density of recruits was almost twenty times higher in the plots of the sub-humid precipitation category than in those of the semi-arid category, 974.0 plants ha$^{-1}$ and 52.6 plants ha$^{-1}$ respectively. The minimum density was very low in both aridity categories, 4.6 and 1.6 plants ha$^{-1}$, respectively, and corresponded to the plots with the highest values of deforestation intensity. Among the recruits in the semi-arid, 22 were saplings (15 years old or younger) and 159 juveniles (16 to 50 years old), compared with 592 saplings and 924 juveniles in the sub-humid (See S3 Table for detailed information). By examining the age structure in more detail, by 10-year cohorts, recruitment patterns differed markedly between precipitation levels (Fig 1). In the semi-arid, the less numerous cohorts were the youngest classes (up to 20 years old) and the most numerous was the 31–40-year-old class. On the contrary, in the sub-humid, the least numerous cohort was the oldest class (41–50 year-old) and the most numerous was the 11–20 year-old class.

### Effect of precipitation level and deforestation intensity on recruitment

Our data show that there is a negative relationship between the intensity of deforestation in the past and the number of recruits per plot at both levels of precipitation (Fig 2). The data also indicate that even at low deforestation level, the number of recruits is very small. For saplings, the effect of past deforestation intensity on the number of recruits differed between precipitation levels, as indicated by the significant interaction term (p-value < 0.0001, Table 1). In the sub-humid, the number of saplings per plot was very high at the minimum deforestation level, but it halved for each 0.1 unit increase in deforestation intensity (95% confidence interval (CI), 0.47 to 0.55). In the semi-arid, however, the number of saplings per plot was so low in all plots that there was no evidence of an effect of past deforestation intensity (p-value = 0.87, Table 1; Fig 2A).

In contrast, for saplings, the interaction between precipitation and deforestation was not significant (p-value = 0.59), indicating that the rate of change in the number of juveniles per plot as deforestation intensity increased was the same for both precipitation levels (Fig 2B). Thus, a 0.1 increase in past deforestation intensity was associated with a 0.72-fold (95% CI, 0.69 to 0.75) decrease in the mean number of juveniles in both precipitation levels. Even though the effect of past deforestation intensity on the rate of change of the number of juveniles was not significantly different between precipitation levels, the total number of juveniles per plot differed notably between precipitation levels. On average, plots in the sub-humid had 14 times more juveniles than plots in the semi-arid at comparable values of past deforestation intensity (95% CI, from 3.85 to 50.56) (Table 1).

### Association with nurse plants

We found 17 shrub and tree species in our study area with the potential to act as nurse plants (S2 Table). However, only seven of them, all trees and large-size shrubs, were significantly and positively associated with *Q. ilex* recruits. Species like *Buxus sempervirens*, *Arctostaphyllos uva-ursi*, the very common Leguminosae shrub *Genista scorpius*, and all the other medium and small-size shrubs, were not positively associated with recruits in our study area.

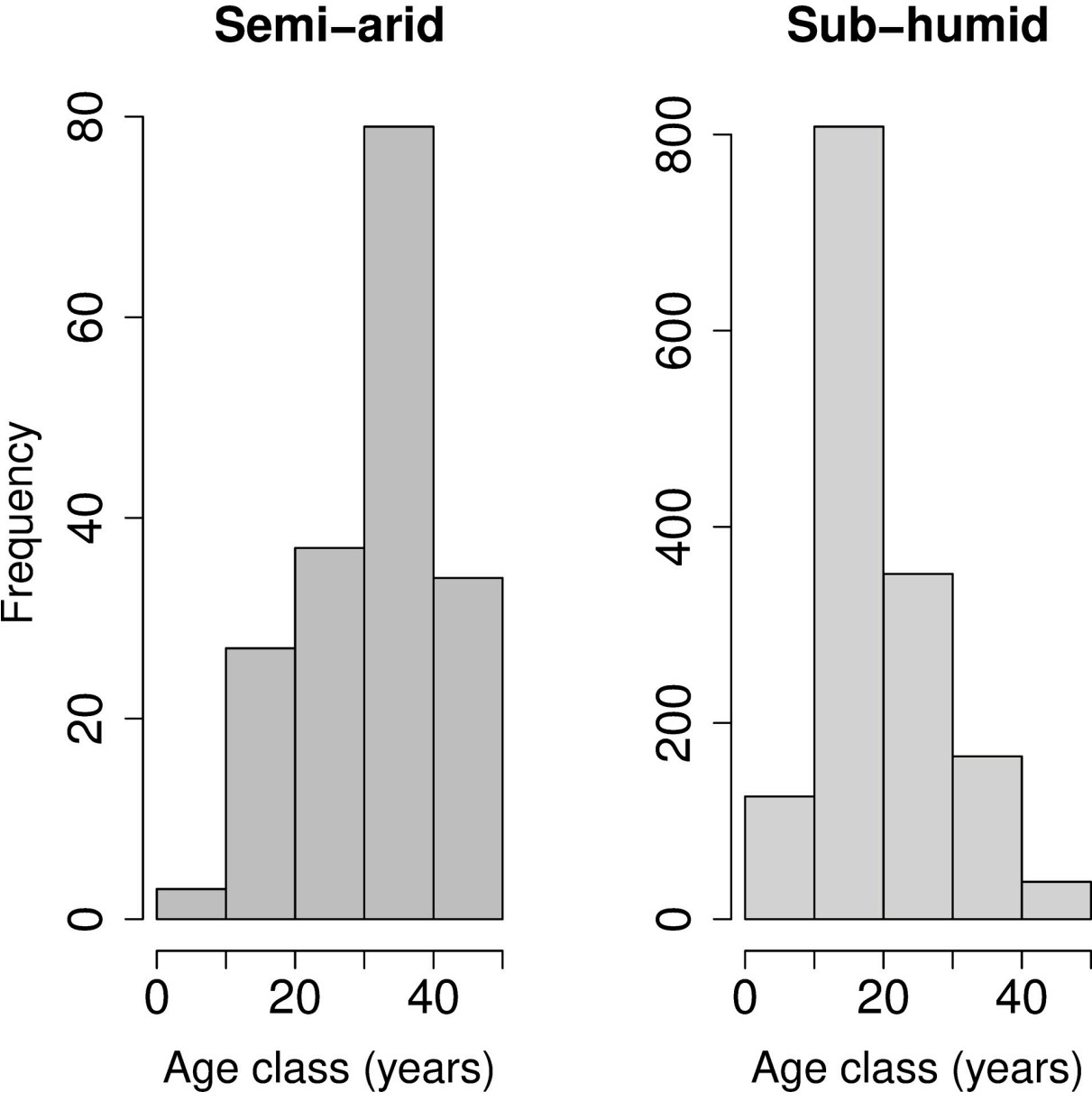

**Fig 1. Age structure of *Quercus ilex* recruits in the two precipitation levels, grouped in 10-year age-classes.**

While most recruits of *Q. ilex* associated with nurse species, there were important differences between age classes of recruits and precipitation levels. In the semi-arid, 20 (91%) of the saplings and 57 (36%) of the juveniles were associated with nurse plants. In the sub-humid, both saplings and juveniles were strongly associated with nurses, 824 (89%) and 569 (96%) respectively. The analysis confirmed the strong association between the presence of saplings and nurse species (p-value < 0.0001; Table 2). The odds of finding a sapling associated with a nurse species were 28.6 times as large as the odds of finding a sapling associated with bare soil and no-nurse plants (95% CI for the odds ratio, between 18.0 and 45.4) and there was no evidence that this odds ratio differed between the semiarid and sub-humid (p-value for the interaction = 0.56).

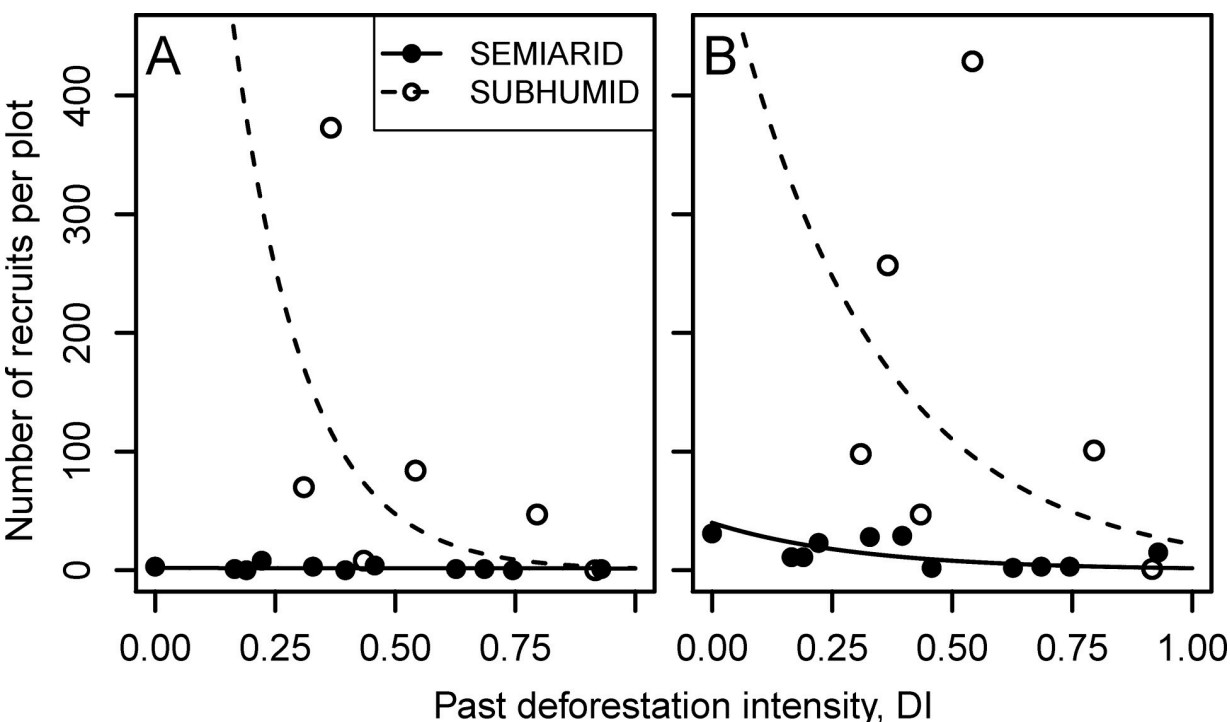

**Fig 2. Effect of intensity of past deforestation on the number of recruits of *Quercus ilex* per plot in each precipitation level.** The lines are the estimated models from the Poisson regression, (A) for saplings (recruits 15 years old or younger) and (B) for juveniles (recruits between 16 and 50 years old).

For juveniles, the association between nurses and recruits differed between semi-arid and sub-humid (p-value for the interaction = 0.0021; Table 2). Juveniles in the semi-arid were not significantly associated to nurses (p-value = 0.15, Table 2). In contrast, the odds of finding a juvenile associated with a nurse species in the sub-humid were 6.7 times as large as the odds of finding a juvenile associated with bare soil and no-nurse plants (95% CI for the odds ratio, between 3.4 and 12.9).

Hierarchical analyses show that none of the examined characteristics of the plot (past deforestation intensity, relative cover of reproductive *Q. ilex* and nurse plants, soil availability and

**Table 1. Parameter estimates from the best fitting Poisson regression models of the effect of precipitation level and intensity of past deforestation on the number of recruits of *Quercus ilex* per plot.**

| Saplings (1–15 years old) | | | |
|---|---|---|---|
| | **Estimate** | **Standard Error** | **p-value** |
| Intercept | 0.5718 | 0.7250 | 0.4290 |
| Precipitation level (sub-humid) | 6.6533 | 1.0291 | <0.0001 |
| Deforestation intensity | -0.2166 | 1.2904 | 0.8670 |
| Precipitation level: Deforestation intensity | -6.5162 | 1.3398 | <0.0001 |
| Juveniles (16–50 years old) | | | |
| | Estimate | Standard Error | p-value |
| Intercept | 3.6856 | 0.3500 | <0.0001 |
| Precipitation level (sub-humid) | 2.6361 | 0.5907 | <0.0001 |
| Deforestation intensity | -3.2351 | 0.1802 | <0.0001 |

**Table 2. Model summaries of best logistic model (GLMM) for the association of recruits of *Quercus ilex* to nurse plants.**

| Saplings (1–15 years old) | | | |
|---|---|---|---|
| | Estimate | Standard Error | p-value |
| Intercept | -6.1466 | 0.5277 | <0.0001 |
| Nurse plants | 3.3544 | 0.2356 | <0.0001 |
| Precipitation level (sub-humid) | 2.6306 | 0.7220 | 0.0003 |
| Juveniles (16–50 years old) | | | |
| | Estimate | Standard Error | p-value |
| Intercept | -2.4085 | 0.3464 | <0.0001 |
| Nurse plants | 0.4721 | 0.3274 | 0.1493 |
| Precipitation level (sub-humid) | 0.8917 | 0.5737 | 0.1201 |
| Precipitation level: Nurse plants | 1.4263 | 0.4636 | 0.0021 |

density of herbivores) affect the strength of the association between recruits and nurse plants (Table 3). This result persists even after including precipitation level in the models. As an exception, there is a significant and positive interaction (p-value = 0.0013) between the presence of nurses at the microsite and the relative cover of nurses in the plot for juvenile plants. In other words, there are relatively more juveniles associated with nurses as the cover of nurses in the plot increases. There is also moderate evidence that herbivory affects the association between juveniles and nurses (p-value = 0.0636), suggesting that increasing herbivory tends to decrease the strength of this association.

## Discussion

The age structure of the population indicates that during the last thirty years, there was a strong reduction in the recruitment of *Q. ilex* in the semi-arid, being very accentuated in the last decade, but in the sub-humid, this reduction only occurred in the last ten years.

As expected both, precipitation level and the intensity of past deforestation, strongly affected the recruitment of *Q. ilex*. The highest recruit density in the sub-humid population

**Table 3. Estimates of the interaction parameter (β11, S3 Appendix) between the indicator of the presence of nurse plants at the microsite level and the plot-level variables.**

| Saplings (1–15 years old) | | | |
|---|---|---|---|
| | Estimate | Standard Error | p-value |
| Deforestation intensity | -0.2678 | 1.2562 | 0.8310 |
| Availability of reproductive *Q. ilex* | 0.0262 | 1.7793 | 0.9880 |
| Nurse availability | 0.5495 | 1.9175 | 0.7744 |
| Soil availability | -0.1577 | 4.0173 | 0.9687 |
| Abundance herbivores | -1.3560 | 1.3450 | 0.3134 |
| Juveniles (16–50 years old) | | | |
| | Estimate | Standard Error | p-value |
| Deforestation intensity | -0.7507 | 1.2741 | 0.5557 |
| Availability of reproductive *Q. ilex* | 1.1988 | 2.1334 | 0.5740 |
| Nurse availability | 5.5797 | 1.7323 | 0.0013 |
| Soil availability | 4.2820 | 3.5760 | 0.2312 |
| Abundance herbivores | -2.4172 | 1.3033 | 0.0636 |

Lack of significance implies that the plot-level variable under consideration did not modify the strength of the association between recruits of *Quercus ilex* and presence of nurse plants.

was almost twenty times more than in the semi-arid population, and in both populations, density decreased rapidly as deforestation intensity increased.

In the semi-arid population, recruit density was very low across all levels of deforestation, less than 51 plants ha$^{-1}$. The increase in deforestation intensity imposed only an additional weak limitation on the density of juveniles (recruits aged 16 to 50), although it had no discernible influence on the density of saplings (recruits aged 1 to 15 years) in these populations.

Our results are consistent with the reported generalized dependence on facilitation of recruitment in *Quercus* species in ecosystems stressed by climate and herbivores [8, 10, 13, 14, 18]. Although we did not studied the facilitation mechanisms of nurses, our results showed that positive association of recruits with nurse plants exists, and that this association was stronger for saplings than for juveniles, as occurs in other species [13]. In fact, we found no evidence of an association between nurses and juveniles in the semi-arid population.

Plot scale factors, such as abundance of nurses, herbivory, soil depth, etc. did not modified the strength of the association between saplings and nurses, reinforcing that is the microsite effect produced by nurses against seed and seedling consumers and providing shade who explain the recruitment pattern. However, for juveniles, the cover of nurses present in the plot positively interact on the association between recruits and nurses, perhaps as a consequence of the correlated positive effect that rural abandonment had on the recovery of all species during the last decades, including both, *Q. ilex* and nurse species.

The reduction in recruitment of *Q. ilex* in the last decades occurred despite the significant decrease in human population and livestock pressure (S1 Appendix), which lead to an increase in both forest species recruitment and forest cover in the western Mediterranean [26, 37]. Average daily temperature between 1901 and 1949 increased at a rate of 0.13˚C per decade, accelerating to more than 0.45˚C per decade since the 1970s to 2012 [3, 32], which includes the recruitment period analysed in this study. The rise in maximum temperatures in mainland Spain from 1951 to 2010 happened mostly in late winter, early spring and summer, while minimum temperature increased in summer, spring and autumn, all the seasons that are most sensitive for photosynthesis, reproduction, growth and seedling survival in this species [38, 39]. Consistently, it has been identified a prolonged period of canopy defoliation of *Q. ilex* and many other forest species across the Iberian Peninsula between 1987 and 2007 [40], which could have also negatively affected the recruitment of *Q. ilex*.

In the semi-arid populations, the low overall low recruitment of *Q. ilex* occurred regardless of the degree of deforestation, suggesting that common mechanisms operated on all sites of this climate category, such as reduced seed production, scarcity of nurse plants, high acorn predation, or limitations to seed germination and seedling survival under those precipitation conditions. On the one hand, seed production in *Q. ilex* is strongly dependent upon water availability in spring and summer [41], and water deficit in the entire region increased along the last decades because the decrease in precipitation and the increase of temperatures [4]. In an accompanying study, P. Garcia-Fayos followed acorn production for ten years (2004–2013) in 150 adults in a *Q. ilex* population located in the semi-arid sites of Santa Eulalia and Cella. During the entire period, seed production was very low. In a scale ranging from complete lack of fruit (0) to mast-fruiting (5), the maximum value assigned to a tree in any given year was 3, while the maximum average value for the population was only 0.9. Low acorn production associates to high rates of predation [12] and, on the other hand, acorn survival is higher under the protection of shrubs and tree species than under the canopy of adults of *Q. ilex* [7, 8]. In our study, we only found two species positively associated to recruits in the semi-arid, and they appeared in a very low frequency, less than half of the plots, and never exceeding 15% of coverage. In contrast, in the sub-humid, we found up to six species potentially acting as nurses, which were present in all plots with a cover equal to or exceeding that of the semi-arid plots (15–39%).

Consequently, the increase in aridity experienced in the region during the last decades [3, 4] suppose a serious problem for the regeneration of the populations of *Q. ilex* in the semi-arid edge of its territory. It is not saying that recruitment of this species is impossible in the semi-arid edge. Models using field evidence of recruitment of *Quercus* species in California, showed an important probability of successful recruitment events after events of several consecutive years wetter than average, even under scenarios of strong aridity [42]. In the semi-arid area studied, without significant regeneration, the current populations of *Q. ilex* can become relict forests, in which long-established trees will be able to survive for some time longer thanks to their ability to sprout after repeated disturbances. In the long term, however, if these adult plants do not have enough time to replenish their carbohydrate stores between disturbance events [43, 44], tree density will decrease and populations will eventually disappear. In sub-humid populations, where conditions are still favorable for recruitment, the increase in temperature and decrease in rainfall predicted by climate change models [45] could also put their future at risk. All of these results highlight the need to incorporate demographic processes, such as recruitment, and field-based evidence into models to predict the effects of climate change on species distribution [42, 46].

In conclusion, the apparent long-term stability of adults of long-lived species that can regrow after stress and disturbance events should not hide that, in some cases, climate change and increased aridity may prevent the recruitment of new individuals. The scarcity of regeneration questions the future of these populations, since the number of individuals will decrease over time, while the ability to adapt to changes will also decrease, since there will be no incorporation of new genetic recombinations in them.

## Supporting information

**S1 Table. Geographical and environmental characteristics of the study plots.** MAP = Mean Annual Precipitation, PET = Potential Evapotranspiration and MAT = Mean Annual Temperature. Plot identification codes follow Moreno-de-las-Heras et al. 2018. Ecosystems, 21: 1295–1305.
(DOCX)

**S2 Table. List of plant species with potential to act as nurse for *Quercus ilex* recruits in our study area.** Bold letters indicate that we found association of that species with recruits of Q. ilex in our study area. Life form: T = tree, LS = large size shrub, MS = medium size shrub, SS = small size shrub and PS = prostrate shrub. Precipitation level: SA = semi-arid, SH = sub-humid, B = both. (*) Several species of the sectio *Rosa caninae*.
(DOCX)

**S3 Table. Summary of the measured variables in each plot.** We consider saplings those recruits aged 15 and younger, and juveniles those aged 16 to 50 years. The values of intensity of past deforestation, area occupied by reproductive *Quercus ilex* and nurses, soil availability and herbivory pressure are ratios. The intensity of past deforestation indicates the loss of tree cover relative to the maximum tree cover in each precipitation level. The area of reproductive *Q. ilex* and nurse plants is the proportion of the area of the plot covered by the canopy of those species. The availability of soil is the proportion of points in the plot where an acorn could germinate and take root. Herbivore pressure is the proportion of subplots in each plot that showed signs of the presence of domestic and wild herbivores (See methods for a detailed variable description).
(DOCX)

**S1 Appendix. Comparison of the abundance of herbivores at the study sites during the last century and at the time of sampling.**
(DOCX)

**S2 Appendix. Modelling the relationships between the estimated age from ring counts and root collar diameter in the field of *Quercus ilex* recruits.**
(DOCX)

**S3 Appendix. Modelling Logistic hierarchical models (microsite models).**
(DOCX)

# Acknowledgments

We want to thank Beatriz López-Gurillo for her enthusiastic and warm help with fieldwork. We are grateful to Manuel García-Hidalgo and Txemi Olano, of the Botany Laboratory of the University of Valladolid in Soria, for their aid in plant age determination and Juan Jiménez, head of the Servei de Vida Silvestre (Generalitat Valenciana, Spain), for kindly providing information on wild herbivore abundance in Vilafranca. The National Meteorological Agency (AEMET, Spain) kindly provided the climatic data of the meteorological stations. The manuscript benefited from discussions within the seminar of the Department of Ecology of CIDE and from the comments of reviewers.

# Author Contributions

**Conceptualization:** Patricio Garcia-Fayos, Esther Bochet.

**Formal analysis:** Patricio Garcia-Fayos, Vicente J. Monleon.

**Funding acquisition:** Patricio Garcia-Fayos, Esther Bochet.

**Investigation:** Patricio Garcia-Fayos, Tiscar Espigares, Jose M. Nicolau, Esther Bochet.

**Methodology:** Patricio Garcia-Fayos, Vicente J. Monleon, Esther Bochet.

**Project administration:** Esther Bochet.

**Writing – original draft:** Patricio Garcia-Fayos.

**Writing – review & editing:** Patricio Garcia-Fayos, Vicente J. Monleon, Tiscar Espigares, Jose M. Nicolau, Esther Bochet.

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
