## [Decision Letter · Decision Letter 0]

13 Aug 2020

PONE-D-20-20482

Aridity threatens the sexual regeneration of Quercus ilex (holm oak) in Mediterranean ecosystems

PLOS ONE

Dear Dr. Garcia-Fayos,

Thank you for submitting your manuscript to PLOS ONE. After careful consideration, we feel that it has merit but does not fully meet PLOS ONE’s publication criteria as it currently stands. Therefore, we invite you to submit a revised version of the manuscript that addresses the points raised during the review process.

Your manuscript was revised by two experts, who have thoughtful comments. Both raised relevant conceptual and methodological concerns that need to be clarified. I agree with their assessment and am willing to consider a revised version for publication in the journal assuming that you modify the manuscript according to the recommendations. 

We look forward to receiving your revised manuscript.

Kind regards,

Angelina Martínez-Yrízar

Academic Editor

PLOS ONE

Journal Requirements:

2. In your Methods section, please provide additional location information of the study sites, including geographic coordinates for the data set if available.

Reviewers' comments:

Reviewer's Responses to Questions

**Comments to the Author**

1. Is the manuscript technically sound, and do the data support the conclusions?

Reviewer #1: Partly

Reviewer #2: Yes

2. Has the statistical analysis been performed appropriately and rigorously? 

Reviewer #1: Yes

Reviewer #2: Yes

3. Have the authors made all data underlying the findings in their manuscript fully available?

Reviewer #1: Yes

Reviewer #2: Yes

4. Is the manuscript presented in an intelligible fashion and written in standard English?

Reviewer #1: Yes

Reviewer #2: Yes

5. Review Comments to the Author

Reviewer #1: General Comments

There is an enormous literature on factors controlling oak regeneration, including oaks in Mediterranean climates of southern Europe and California. Quercus ilex is especially well-studied using both experimental and survey methods. This paper presents a survey-based case study of oak regeneration in the Iberian Range in Spain. The authors compare 11 plots in a relatively dry “semi-arid” area to 6 plots in a “sub-humid area. Plots are surveyed only once to census adult, sapling, and seedling oaks. Fairly simple or indirect proxies are used to characterize past land use history, ungulate activity, nurse plant effect, and soil seed bed quality. Ages of 1697 seedling and sapling recruits is estimated based on the relationship between plant age and root collar diameter (r^2 =0.48 and r^2 = 0.65 for semi-arid and sub-humid sites, respectively). The authors document a reduction of holm oak recruitment over the past 30 years at the semi-arid plots and over the past 10 years at the sub-humid plots. They ascribe recent recruitment declines mainly to increasing aridity.

Surveys such as this require much time and effort and I commend the authors for their care in executing the work. Their conclusions are weakened by small sample size, sketchy information on site characteristics and site history, reliance on proxies for seed supply and herbivore pressure, and absence of formal experimental manipulations. I find the most interesting results to be the differences in results for seedling vs. sapling phases , which are supported by better (although still only moderately reliable) age estimates than most studies.

The authors make practically no effort to relate the results for their holm oak survey to the large international literature on other oak species in California and Europe. PLoS has an international readership and most readers will not be especially interested in a correlational study of a single species in one region unless more effort is made to compare the findings to closely related work on oak recruitment, nurse plants, herbivory, and climate change, in other areas and for similar species. Thus my major recommendation to the authors is that they better contextualize their work in the introduction and discussion sections in the context of what we know and don’t know about oak regeneration in Mediterranean climates and how their results contribute to a large and growing literature on the interaction of factors affecting oak population dynamics under rapid climate change. The discussion is already very long; I would advise tightening and shortening the text devoted to your study sites and species and make more of an effort to relate your work to the larger literature.

Other comments:

Lines 40-47 This paragraph is pretty basic and in my opinion unnecessary.

l. 62 I would replace “model species” with “case study”

l. 117 “repeatedly in the past” is vague. Is historical aerial photography available to reconstruct recent historic tree cover change at these sites? You use the term “intensity of past deforestation” through the paper but your observed variable is tree cover at the time of the survey relative to expected maximum cover at that rainfall level. Without knowing more about the changes through time in tree cover as a result of land use activities, you cannot relate recruitment dynamics over the past half century directly to “intensity of past deforestation”, only to today’s relative tree cover. I would use different terminology.

L 177. Why did you use mean diameter instead of equivalent diameter? The perpendicular measurements are typically used to characterize the long axis and short axis of an ellipse from which an equivalent diameter of a circle can be estimated.

L 178. You are not able to study the “ direct effect of nurse plants”, only the association of recruits with shrubs.

L180. “Nurse plant” is a loaded term that implies that the shrubs were there and presumably well-established before the seedling or sapling recruited. Is this a reasonable assumption in all cases? There is no way to establish cause and effect here, only spatial association of recruits and shrubs. Whether the shrubs actually served as nurse plants is conjectural (albeit a reasonable conjecture).

L 189. Did you test whether the relationship between size and age varied for recruits associated with nurse plants versus those without?

L 204-205. Given that you are assigning plants to age classes based on size, you should report the classification accuracy, rather than the squared correlation between root collar diameter and age (line 226). That includes classifying plants as <15,16-50, and >50 years, as well as for classifying plants into 10-year age classes.

L 228-230. I would report mean densities rather than total number of recruits here. And Table 1, which simply reports the raw data, including counts, could be a supplementary table.

L 235-238. These conclusions depend on the accuracy of age class assignments.

L 255-258. These results are difficult to interpret given that height reflects both browsing pressure and site productivity. If you think this difference in height is important you should probably undertake a more formal statistical comparison of height distributions.

L 402. You have not established that these species “acted as nurses.”

L. 410-426. Even under a warming-drying trend, oak establishment can still occur in unusually wet years (e.g. Serra-Diaz et al., 10.1111/ecog.02074). You have pretty limited evidence that the current pattern of size/age of seedlings and saplings in your plots is a direct consequence of climate change. I would be more circumspect in discussing this demographic pattern and its interpretation.

Reviewer #2: This study deals with the regeneration ecology of holm oak (Quercus ilex), a keystone tree in Mediterranean forests. The main dataset is the inventory of 1697 recruits, distributed across 17 sites with contrasted amount of annual rainfall: 11 sites with 400-450 mm (semiarid) versus 6 sites with 600-650 mm (subhumid). In each climatic type, a gradient of densities of oak trees was selected, assuming that it reflected the (inverted) gradient of past deforestation pressure. The age of each recruit was estimated, based on tree-ring counting of a sample of 171 recruits and a model of the age-diameter relationships.

A main finding was that in semiarid sites recruit density was very low (16 times less than in subhumid) and the youngest age class (<20 years old) had the lowest frequency. All these results indicating a lack of regeneration and questioning the viability of those populations under semiarid conditions.

In addition, authors analysed the association between oak recruits and nurse plants, discussing the relevance of facilitation for the successful oak recruitment under semiarid conditions.

This study is based on extensive field work to quantify oak recruitment and on development of models to explain their relationships with several factors at plot scale: annual rainfall, tree density (surrogate of past deforestation), effects of nurse plants, and livestock density (surrogate of herbivory pressure). The results will interest to ecologists and forest managers, in particular those concerned with Mediterranean oak forests.

Some particular comments and suggestions are following:

Title: perhaps it would be more adequate: “Increasing aridity threatens…” to stress the effects of climate change (not just aridity). As repeated several times in the manuscript, the “increased” aridity matters; e.g., L. 30-31, L. 405-406, L. 485-486.

L. 83. Authors stated that “we analysed the recruitment of Q. ilex over the last 50 years…”. This phrase seems ambiguous, because this is not a long-term study monitoring recruitment during 50 years. It should be clarified.

L. 88-89. Authors assumed that in the studied semiarid sites “the species was on the edge of the precipitation niche.” That assumption would be according to the present distribution. Is possible that holm oak forests could occupy drier areas but they were eliminated by historical deforestation? Any evidence about the historical “edge” of the species distribution?

L. 110-111. The statement “environmental characteristics, vegetation type and human use were similar in all plots” seems an assumption for statistical purpose. However, it is difficult to imagine that similarity for 17 sites across a 130 km transect, given the heterogeneity in natural conditions and land use, typical of Mediterranean landscapes. Perhaps, it could be clarified the sentence.

L. 114. What are “high areas”? Those with high elevation?

L.132-137. The current density of oak trees is used as surrogate (inverse) of past deforestation. It is assumed that in historical times all the sites had 100% Q. ilex and that current density is result of deforestation. How to rule out that sites on the distribution edge are not at colonizing stage? That is, with low density but without previous deforestation.

On the other hand, in subhumid sites frequently Q. ilex is mixed with Q. faginea (see S2 Appendix). In those mixed forests, after disturbance Q. ilex density can increase replacing Q. faginea. Were those mixed oak forests excluded from the 6 subhumid sites?

L. 138. Please clarify why 583 plots (374 plus 209) were used to calculate deforestation index? Were the 17 studied sites included in that wider dataset?

L. 107, L. 138-139. Apparently, plot selection and deforestation index were based on a previous work by Moreno de las Heras et al. (2018) (reference 27). What is the relation with the present study? Perhaps it should be made explicit that relation in the Introduction and Discussion sections.

L. 157-159. Authors spent extensive effort to evaluate the herbivory pressure on oak recruits by indirect evidences (faeces, footprints, hairs, etc.) in each plot. Was there any “direct” evidence of browsing on the 1697 tallied recruits?

L. 164. Recruits were defined as young oaks with base diameter less than 50mm; once measured, they had estimated ages from 2 to 50 years. L. 203-205. Based on previous studies of ontogenetic shift in the response to shade, they were grouped in 2 classes: 1-15 years and 16-50 years, in order to analyse the response to the gradient of tree density/deforestation (Fig. 2). The 2 age classes were called “seedlings” and “saplings”, which is somewhat confusing. Although “seedling” is any plant grown from a seed, in demographic studies “seedlings” refer usually to young plants still depending on seed reserves, that is 1-3 years in oaks. A suggestion could be call them “saplings” (2-15 years) and “juveniles” (16-50 years), or just 1st stage and 2nd stage recruits. Also is surprising to know about “saplings” of 50 years, when the usual maturity of Q. ilex for acorn production is 15-20 years. Probably they are supressed by coexisting adults as a “juvenile” bank.

L. 261-262, L. 296-301. There are too few points in figure 2 (6 points for the subhumid model) to infer that “the intensity of past deforestation had a strong negative influence on the number of recruits…” Perhaps a more cautious writing is recommended. On the other hand, we can interpret the same figure as a very low recruitment in the almost treeless plots (DI near 1), a logical consequence of very low acorn production by only a few trees.

L. 389-391, L.398-399. This is an interesting discussion on the multistage limitations of oak recruitment. However, some discussion on the stage of seed predation is missing; that demographic stage has been revealed as critical for oak regeneration under Mediterranean conditions (e.g., Pulido & Diaz, 2005, Ecoscience 12: 92-102).

L. 470-473. Interesting point about the “persistence niche”. What would be the maximum life-span of Quercus ilex in those sites? Several hundred of years? What is the probability of changing the environmental conditions during that period?

L. 507. The list of references is complete and useful for readers interested on oak regeneration and climate change.

6. PLOS authors have the option to publish the peer review history of their article (what does this mean?). If published, this will include your full peer review and any attached files.

Reviewer #1: No

Reviewer #2: No

---

## [Author Response · Author response to Decision Letter 0]

10 Sep 2020

Yes, the new manuscript meets those requirements.

2. In your Methods section, please provide additional location information of the study sites, including geographic coordinates for the data set if available.

We have now included this information of the plots in the form of a new table in supplementary material (S1 Table)

Permits were not required to sample the study plots since they were not located in areas with restricted access and the type of sampling was neither extractive nor destructive, except in the case of Q. ilex individuals necessary to establish age relationships with diameter. 

The collection of Q. ilex is not subject to any prohibition by laws or regulations of a national, regional or local nature that affect the study territory except for the use of firewood, which, given the size of the individuals sampled for this research, it would not apply.

The data files have been deposited in the Figshare repository (DOI 10.6084 / m9.figshare.12855170) and we will make them public once the article is accepted for publication by the journal. You and the reviewers can access them at https://figshare.com/s/e497b238595e190548a5

Done

Reviewers' comments:

5. Review Comments to the Author

Reviewer #1: General Comments

There is an enormous literature on factors controlling oak regeneration, including oaks in Mediterranean climates of southern Europe and California. Quercus ilex is especially well-studied using both experimental and survey methods. This paper presents a survey-based case study of oak regeneration in the Iberian Range in Spain. The authors compare 11 plots in a relatively dry “semi-arid” area to 6 plots in a “sub-humid area. Plots are surveyed only once to census adult, sapling, and seedling oaks. Fairly simple or indirect proxies are used to characterize past land use history, ungulate activity, nurse plant effect, and soil seed bed quality. Ages of 1697 seedling and sapling recruits is estimated based on the relationship between plant age and root collar diameter (r^2 =0.48 and r^2 = 0.65 for semi-arid and sub-humid sites, respectively). The authors document a reduction of holm oak recruitment over the past 30 years at the semi-arid plots and over the past 10 years at the sub-humid plots. They ascribe recent recruitment declines mainly to increasing aridity.

Surveys such as this require much time and effort and I commend the authors for their care in executing the work. Their conclusions are weakened by small sample size, sketchy information on site characteristics and site history, reliance on proxies for seed supply and herbivore pressure, and absence of formal experimental manipulations. I find the most interesting results to be the differences in results for seedling vs. sapling phases , which are supported by better (although still only moderately reliable) age estimates than most studies.

The authors make practically no effort to relate the results for their holm oak survey to the large international literature on other oak species in California and Europe. PLoS has an international readership and most readers will not be especially interested in a correlational study of a single species in one region unless more effort is made to compare the findings to closely related work on oak recruitment, nurse plants, herbivory, and climate change, in other areas and for similar species. Thus my major recommendation to the authors is that they better contextualize their work in the introduction and discussion sections in the context of what we know and don’t know about oak regeneration in Mediterranean climates and how their results contribute to a large and growing literature on the interaction of factors affecting oak population dynamics under rapid climate change. The discussion is already very long; I would advise tightening and shortening the text devoted to your study sites and species and make more of an effort to relate your work to the larger literature.

The new version of the manuscript includes more information on the environmental characteristics of all sampled plots, we also added a new Appendix (S1 Table). We have also expanded the information about the experimental plots and their history of use in the manuscript (Lines 98-124 in the “Revised Manuscript with Track Changes” file) to increase the reliability of the proxies used for some of the variables as well as the confidence that the plots are homogeneous in those factors other than the ones we wanted to study. Despite this, we control the writing so as not to substantially modify the length of the manuscript.

For clarity, below we justify the reasons for our choice of some of the proxies used and for which it is not possible to find other more exact indicators:

-As there are no records of acorn production and Quercus ilex is a recognized masting species, it is difficult to have accurate data on the quantity of seeds produced at each site during the years involved in this study. Therefore, we consider that the best proxy of this quantity that we could have is the abundance of reproductive plants of this species in each plot.

-There are no precise censuses of wild herbivores in the study areas for the period studied. In addition, both domestic and wild herbivores roam wide areas throughout each day and move between areas throughout the seasons, but use those that are most favorable to them more intensively. For this reason, we consider that a proxy for the intensity with which herbivores use each plot is the frequency with which traces of their activity are found, including those that have a relatively long life (more than one year), such as excrement, hair and wool. 

-The evolution of the number of heads of domestic livestock from the mid-19th and early 20th centuries, when the highest human population density was reached in the region, to the present day after rural abandonment, allows us to consider with sufficient confidence that the density of domestic cattle cannot be considered in any case as overgrazing during the studied recruitment period, and that it remains reasonably homogeneous between the different study areas. Likewise, we also are confident that this decrease in density has not been compensated by the increase in density of wild herbivores.

Other comments:

Lines 40-47 This paragraph is pretty basic and in my opinion unnecessary

We deleted it.

l. 62 I would replace “model species” with “case study”

Since we have modified the Introduction to accommodate your suggestion to expand the context to other oaks and to include suggestions that reviewer # 2 made, this sentence has not been included in the new version.

l. 117 “repeatedly in the past” is vague. Is historical aerial photography available to reconstruct recent historic tree cover change at these sites? You use the term “intensity of past deforestation” through the paper but your observed variable is tree cover at the time of the survey relative to expected maximum cover at that rainfall level. Without knowing more about the changes through time in tree cover as a result of land use activities, you cannot relate recruitment dynamics over the past half century directly to “intensity of past deforestation”, only to today’s relative tree cover. I would use different terminology

In accordance with this suggestion and other from reviewer#2 [L. 107, L. 138-139], we have modified the entire section in such a way as to clarify as much as possible the doubts raised about the interpretation of the deforestation values used in this study and the underlying causes of variation in Q. ilex cover (see Lines 110-124 in the “Revised Manuscript with Track Changes” file). 

L 177. Why did you use mean diameter instead of equivalent diameter? The perpendicular measurements are typically used to characterize the long axis and short axis of an ellipse from which an equivalent diameter of a circle can be estimated

It is true that the great irregularity of shapes of the section of the root collar of the recruits allows the use of different parameters, such as the equivalent diameter indicated by the reviewer, or others such as the mean of the major and minor axes and that of the orthogonal axes. However, since from the first moment, we tried to model the age of the plants from the basal area, and this is determined from the orthogonal diameters measured with calipers, we decided to use the mean diameter as a parameter.

L 178. You are not able to study the “direct effect of nurse plants”, only the association of recruits with shrubs

L180. “Nurse plant” is a loaded term that implies that the shrubs were there and presumably well-established before the seedling or sapling recruited. Is this a reasonable assumption in all cases? There is no way to establish cause and effect here, only spatial association of recruits and shrubs. Whether the shrubs actually served as nurse plants is conjectural (albeit a reasonable conjecture)

(this is our answer to the two previous comments): It is true. Consequently, we have modified the wording of this sentence, as well as the allusion to this effect in other places in the manuscript (Lines 88-89 in the “Revised Manuscript with Track Changes” file).

L 189. Did you test whether the relationship between size and age varied for recruits associated with nurse plants versus those without?

No, we have not analyzed it in this manuscript. Its analysis would add complexity and we believe that it is not essential for the discourse we intend. In any case, we welcome the suggestion, since this and other aspects of the association with nurses we intend to develop them in a future manuscript.

L 204-205. Given that you are assigning plants to age classes based on size, you should report the classification accuracy, rather than the squared correlation between root collar diameter and age (line 226). That includes classifying plants as <15,16-50, and >50 years, as well as for classifying plants into 10-year age classes

We have performed precision analyses of the classification of seedlings into the age classes that we used in our study. The results and conclusions of these analyses have been included in the Appendix S4. We do not apply this analysis for the accuracy of the classification in the 10-year age classes of Figure 1 because this figure is used and discussed as a descriptive question of the distribution of plants in the population, not a formal analysis due to the small sample size.

L 228-230. I would report mean densities rather than total number of recruits here. And Table 1, which simply reports the raw data, including counts, could be a supplementary table

We have now included information on recruit density in the text (Lines 242-246 in the “Revised Manuscript with Track Changes” file) and have moved the table with the raw data to Appendix S6 in the supplementary material, as suggested.

L 235-238. These conclusions depend on the accuracy of age class assignments

It is true. See our answer to the question posed in L 204-205 above.

L 255-258. These results are difficult to interpret given that height reflects both browsing pressure and site productivity. If you think this difference in height is important you should probably undertake a more formal statistical comparison of height distributions

It is true. Now, since we have no way to assign the differences in the height of the plant to one or the other factor, we have decided to delete this paragraph

L 402. You have not established that these species “acted as nurses.”

Ok, see our response to your previous comments (L178 and L 180) on this topic.

L. 410-426. Even under a warming-drying trend, oak establishment can still occur in unusually wet years (e.g. Serra-Diaz et al., 10.1111/ecog.02074). You have pretty limited evidence that the current pattern of size/age of seedlings and saplings in your plots is a direct consequence of climate change. I would be more circumspect in discussing this demographic pattern and its interpretation

Is right. In the new version of the manuscript, we have changed most of the discussion to highlight what evidence we really have and be more cautious in its interpretation (Lines 361-405 in the “Revised Manuscript with Track Changes” file). In addition, we have added a specific comment to the fact that it is possible that sufficient recruitment can occur despite the increase in aridity when there are several consecutive years with suitable conditions, which is still possible (lines 393-396 in the “Revised Manuscript with Track Changes” file).

Reviewer #2: This study deals with the regeneration ecology of holm oak (Quercus ilex), a keystone tree in Mediterranean forests. The main dataset is the inventory of 1697 recruits, distributed across 17 sites with contrasted amount of annual rainfall: 11 sites with 400-450 mm (semiarid) versus 6 sites with 600-650 mm (subhumid). In each climatic type, a gradient of densities of oak trees was selected, assuming that it reflected the (inverted) gradient of past deforestation pressure. The age of each recruit was estimated, based on tree-ring counting of a sample of 171 recruits and a model of the age-diameter relationships.

A main finding was that in semiarid sites recruit density was very low (16 times less than in subhumid) and the youngest age class (<20 years old) had the lowest frequency. All these results indicating a lack of regeneration and questioning the viability of those populations under semiarid conditions.

In addition, authors analysed the association between oak recruits and nurse plants, discussing the relevance of facilitation for the successful oak recruitment under semiarid conditions.

This study is based on extensive field work to quantify oak recruitment and on development of models to explain their relationships with several factors at plot scale: annual rainfall, tree density (surrogate of past deforestation), effects of nurse plants, and livestock density (surrogate of herbivory pressure). The results will interest to ecologists and forest managers, in particular those concerned with Mediterranean oak forests.

Some particular comments and suggestions are following:

Title: perhaps it would be more adequate: “Increasing aridity threatens…” to stress the effects of climate change (not just aridity). As repeated several times in the manuscript, the “increased” aridity matters; e.g., L. 30-31, L. 405-406, L. 485-486

The reviewer observation seems very timely to us, so we have modified both the main title and the short title accordingly.

L. 83. Authors stated that “we analysed the recruitment of Q. ilex over the last 50 years…”. This phrase seems ambiguous, because this is not a long-term study monitoring recruitment during 50 years. It should be clarified

We agree with the reviewer's suggestion. Accordingly, we have modified the paragraph to avoid the false idea that we have monitored the recruitment of this species for the last 50 years (see Lines 74-75 in the “Revised Manuscript with Track Changes” file).

L. 88-89. Authors assumed that in the studied semiarid sites “the species was on the edge of the precipitation niche.” That assumption would be according to the present distribution. Is possible that holm oak forests could occupy drier areas but they were eliminated by historical deforestation? Any evidence about the historical “edge” of the species distribution?

In the paper, we have assumed that the species was on the edge of the precipitation niche at the beginning of the studied recruitment period based on the information provided in reference [27] on the climatic niche of the species. In this reference, the authors indicate that they obtained the precipitation niche of Q. ilex for the Iberian Peninsula by crossing the geographical distribution of the species for the year 1985 with the precipitation data from the series of observations 1940-1989. On the other hand, we obtained the precipitation values of the plots used in our study from reference [31], which uses the series of precipitation observations for the period 1951-1999.

In any case, we have added a comment in the manuscript that clarifies this matter (Lines 82-83 in the “Revised Manuscript with Track Changes” file).

L. 110-111. The statement “environmental characteristics, vegetation type and human use were similar in all plots” seems an assumption for statistical purpose. However, it is difficult to imagine that similarity for 17 sites across a 130 km transect, given the heterogeneity in natural conditions and land use, typical of Mediterranean landscapes. Perhaps, it could be clarified the sentence

In order to clarify this statement, we have now added more information on the environmental characteristics of the parcels: see the new wording of the section “Selection and characteristics of the plots” and a new appendix (S1 Table)

L. 114. What are “high areas”? Those with high elevation?

True. Now we have modified this sentence accordingly (Lines 103-104 in the “Revised Manuscript with Track Changes” file).

L.132-137. The current density of oak trees is used as surrogate (inverse) of past deforestation. It is assumed that in historical times all the sites had 100% Q. ilex and that current density is result of deforestation. How to rule out that sites on the distribution edge are not at colonizing stage? That is, with low density but without previous deforestation

We have rewritten everything regarding the study area and the selection of plots (Lines 110-115 and 120-123 in the “Revised Manuscript with Track Changes” file). In the previous version, we directed readers to a previous article of ours in which we described all this in detail. Now, we have included much of this information in the current version of the manuscript to ensure that readers have confidence that all plots are reasonably homogeneous in relation to the factors that can affect colonization and development of Q. ilex, except for the factors included in the experimental design (level of precipitation and intensity of deforestation), and that it is also clear to the reader that the differences between the plots in Q. ilex tree cover are due to human deforestation in the past.

On the other hand, in subhumid sites frequently Q. ilex is mixed with Q. faginea (see S2 Appendix). In those mixed forests, after disturbance Q. ilex density can increase replacing Q. faginea. Were those mixed oak forests excluded from the 6 subhumid sites?

Although we do not rule out that, as the reviewer indicates, it is possible that in the past Q. ilex has been favored to the detriment of Q. faginea, we are confident that this is not the case in any of the selected plots. The forests of Q. faginea in the Iberian Range on calcareous substrate are only found on marls and in deep colluvial soils. In the environmental conditions in which we selected the plots (flat limestone areas with shallow soil), it is not possible to find Q. faginea forests or mixed forests with both species throughout the region. We only found plots where Q. ilex is the dominant tree species, although isolated individuals of Q. faginea sometimes appear.

L. 138. Please clarify why 583 plots (374 plus 209) were used to calculate deforestation index? Were the 17 studied sites included in that wider dataset?

We have included an explanation about it (see Lines 141-144 in the “Revised Manuscript with Track Changes” file)

L. 107, L. 138-139. Apparently, plot selection and deforestation index were based on a previous work by Moreno de las Heras et al. (2018) (reference 27). What is the relation with the present study? Perhaps it should be made explicit that relation in the Introduction and Discussion sections

See our answer to the comment L 117 of the Reviewer #1

L. 157-159. Authors spent extensive effort to evaluate the herbivory pressure on oak recruits by indirect evidences (faeces, footprints, hairs, etc.) in each plot. Was there any “direct” evidence of browsing on the 1697 tallied recruits?

No, we did not observe direct evidence of herbivory in the species in any of the studied sites, neither in adults nor in juveniles or seedlings. Therefore, we decided to use indirect evidence to inform us of the pressure of herbivory. We have now included a sentence (Lines 160-163 in the “Revised Manuscript with Track Changes” file) in the paper to illustrate readers about this reviewer observation.

L. 164. Recruits were defined as young oaks with base diameter less than 50mm; once measured, they had estimated ages from 2 to 50 years. L. 203-205. Based on previous studies of ontogenetic shift in the response to shade, they were grouped in 2 classes: 1-15 years and 16-50 years, in order to analyse the response to the gradient of tree density/deforestation (Fig. 2). The 2 age classes were called “seedlings” and “saplings”, which is somewhat confusing. Although “seedling” is any plant grown from a seed, in demographic studies “seedlings” refer usually to young plants still depending on seed reserves, that is 1-3 years in oaks. A suggestion could be call them “saplings” (2-15 years) and “juveniles” (16-50 years), or just 1st stage and 2nd stage recruits

Agree. We have changed the names as suggested by the reviewer: saplings instead of seedlings and juveniles instead of saplings throughout the main text, tables, figures and supplementary material.

Also is surprising to know about “saplings” of 50 years, when the usual maturity of Q. ilex for acorn production is 15-20 years. Probably they are supressed by coexisting adults as a “juvenile” bank

Yes, it is also what we believe. We were also surprised to find such suppressed individuals, since in the scientific literature we find references to a seedling bank not much more than 15 years old.

L. 261-262, L. 296-301. There are too few points in figure 2 (6 points for the subhumid model) to infer that “the intensity of past deforestation had a strong negative influence on the number of recruits…” Perhaps a more cautious writing is recommended. On the other hand, we can interpret the same figure as a very low recruitment in the almost treeless plots (DI near 1), a logical consequence of very low acorn production by only a few trees

Yes. Consequently, we have now written this paragraph in a more cautious manner (Lines 261-263 in the “Revised Manuscript with Track Changes” file) and examine it in the Discussion section (Lines 337-347 and 373-390 in the “Revised Manuscript with Track Changes” file).

L. 389-391, L.398-399. This is an interesting discussion on the multistage limitations of oak recruitment. However, some discussion on the stage of seed predation is missing; that demographic stage has been revealed as critical for oak regeneration under Mediterranean conditions (e.g., Pulido & Diaz, 2005, Ecoscience 12: 92-102)

We have modified the introduction and discussion to include and further develop the role of predation in recruitment (Lines 55-65 and Lines 354-390 in the “Revised Manuscript with Track Changes” file).

L. 470-473. Interesting point about the “persistence niche”. What would be the maximum life-span of Quercus ilex in those sites? Several hundred of years? What is the probability of changing the environmental conditions during that period?

To our knowledge, there is no scientific publication in which the age that this species can reach is reported. In addition to this, researchers who have attempted to determine age frequently find that the individuals they can sample are actually sprouts from a much older stump, so that sprouts that are 100 years old can very rarely be found. On the other hand, the available climatic reconstructions report only oscillations in recent centuries since at least the end of the 17th century, and not important changes.

L. 507. The list of references is complete and useful for readers interested on oak regeneration and climate change

While revising your submission, please upload your figure files to the Preflight Analysis and Conversion Engine (PACE) digital diagnostic tool, https://pacev2.apexcovantage.com/. PACE helps ensure that figures meet PLOS requirements. Please note that Supporting Information files do not need this step

Done

---

## [Editor Report · Decision Letter 1]

14 Sep 2020

Increasing aridity threatens the sexual regeneration of Quercus ilex (holm oak) in Mediterranean ecosystems

PONE-D-20-20482R1

Dear Dr. Garcia-Fayos,

We’re pleased to inform you that your manuscript has been judged scientifically suitable for publication and will be formally accepted for publication once it meets all outstanding technical requirements.

Kind regards,

Angelina Martínez-Yrízar

Academic Editor

PLOS ONE
---

## [Editor Report · Acceptance letter]

22 Sep 2020

PONE-D-20-20482R1 

Increasing aridity threatens the sexual regeneration of *Quercus*
*ilex*(holm oak) in Mediterranean ecosystems 

Dear Dr. Garcia-Fayos:

I'm pleased to inform you that your manuscript has been deemed suitable for publication in PLOS ONE. Congratulations! Your manuscript is now with our production department. 

Kind regards, 

on behalf of

Dr. Angelina Martínez-Yrízar 

Academic Editor

PLOS ONE